# Personalized Endoscopy in Complex Malignant Hilar Biliary Strictures

**DOI:** 10.3390/jpm11020078

**Published:** 2021-01-29

**Authors:** Ivo Boškoski, Tommaso Schepis, Andrea Tringali, Pietro Familiari, Vincenzo Bove, Fabia Attili, Rosario Landi, Vincenzo Perri, Guido Costamagna

**Affiliations:** 1Center for Endoscopic Research Therapeutics and training (CERTT), Università Cattolica del Sacro Cuore, 20123 Rome, Italy; tommaso.schepis@gmail.com (T.S.); andrea.tringali@unicatt.it (A.T.); pietro.familiari@unicatt.it (P.F.); vincenzo.bove@policlinicogemelli.it (V.B.); fabia.attili@policlinicogemelli.it (F.A.); rosario.landi@policlinicogemelli.it (R.L.); vincenzo.perri@unicatt.it (V.P.); guido.costamagna@unicatt.it (G.C.); 2Digestive Endoscopy Unit, Fondazione Policlinico Universitario Agostino Gemelli IRCCS, 00168 Rome, Italy

**Keywords:** personalized endoscopy, malignant hilar strictures, self-expandable metal stents, plastic biliary stents, endoscopic retrograde cholangiopancreatography, peroral cholangioscopy, confocal laser endomicroscopy, endoscopic ultrasound

## Abstract

Malignant hilar biliary obstruction (HBO) represents a complex clinical condition in terms of diagnosis, surgical and medical treatment, endoscopic approach, and palliation. The main etiology of malignant HBO is hilar cholangiocarcinoma that is considered an aggressive biliary tract’s cancer and has still today a poor prognosis. Endoscopy plays a crucial role in malignant HBO from the diagnosis to the palliation. This technique allows the collection of cytological or histological samples, direct visualization of the suspect malignant tissue, and an echoendoscopic evaluation of the primary tumor and its locoregional staging. Because obstructive jaundice is the most common clinical presentation of malignant HBO, endoscopic biliary drainage, when indicated, is the preferred treatment over the percutaneous approach. Several endoscopic techniques are today available for both the diagnosis and the treatment of biliary obstruction. The choice among them can differ for each clinical scenario. In fact, a personalized endoscopic approach is mandatory in order to perform the proper procedure in the singular patient.

## 1. Introduction

The management of malignant hilar biliary obstruction (HBO) is still today a medical challenge in terms of diagnosis, treatment alternatives, and palliation options. The etiology of malignant HBO includes mainly the cholangiocarcinoma originating between the cystic duct and the segmental branch of the intrahepatic bile ducts (Klatskin tumor or hilar cholangiocarcinoma) [1]. More rarely, hilar obstruction can be determined by the local extension of adjacent tumors, such as gallbladder, liver, and pancreatic cancer, or by metastasis from distant malignancies [2]. The Bismuth-Corlette classification system is used to classify hilar cholangiocarcinoma taking into account the involvement of the biliary confluence and the intrahepatic ducts. Patients may be classified into four categories—Bismuth type I when the stricture is localized in the main biliary duct and does not involve the confluence; type II when the stricture involves the main confluence; type IIIa when the stricture involves the confluence and the right sectorial confluence sparing the left one; type IIIb when the stricture involves the confluence and the left sectorial confluence sparing the right one; and type IV when confluence, right, and left sectorial confluence are all involved [3].

The prognosis of hilar cholangiocarcinoma is poor for the locally aggressive behavior with a tendency to infiltrate adjacent tissues and neural, perineural, and lymphatic involvement [4]. The only curative approach is surgery; however, it is feasible only in 30–40% of patients [5]. The clinical presentation generally consists of obstructive jaundice and cholangitis, however, cholangiocarcinoma can be asymptomatic for a long time [4].

Endoscopy plays a crucial role in terms of diagnosis, management of obstructive jaundice, and palliation. The complexity of malignant HBO makes each patient a singular case requiring a personalized approach from the diagnosis to the final treatment. In this setting, a multidisciplinary is mandatory with the involvement of surgeons, endoscopists, interventional radiologists, and oncologists in order to find the best approach that properly fits with the patients’ characteristics.

The aim of this study was to perform a review of the current literature in order to define the current role of endoscopy in diagnosis, biliary drainage, and palliation of malignant HBO and their application in the various clinical settings, and to assess the role of personalized endoscopy in the treatment of complex malignant hilar biliary strictures.

## 2. Diagnostic Approach in HBO

The diagnostic approach to HBO has the aim to distinguish whether a stricture is malignant or benign and, in case of malignancy, to evaluate the resectability. The patient with HBO generally presents cholestasis and obstructive jaundice, in this setting the first-line imaging technique is classically an abdominal ultrasound (US). Although it is a useful tool in the initial management, US is an operator-dependent technique and is burdened by low sensitivity and specificity [6]. Multidetector computed tomography (CT), magnetic resonance imaging (MRI), and magnetic resonance cholangiopancreatography (MRCP) are generally required to perform an accurate diagnosis. CT and MRCP are complementary—on one hand, CT can provide information on the primary tumor, locoregional infiltration, vascular infiltration, and distant metastasis, and on the other hand, MRCP has the best sensitivity and specificity in defining the intraductal extension [7,8]. The role of ^18^F-fluorodeoxyglucose-positron emission tomography (FDG-PET) is controversial; it can be appropriate in case of indeterminate lesions and in the detection of distant metastasis, however, it is not routinely performed and its application should be evaluated in the singular patient [4].

Endoscopy plays a crucial role in the diagnostic algorithm in patients with malignant HBO mainly for the collection of cytological or histological samples.

Endoscopic Retrograde Cholangio-Pancreatography (ERCP) is nowadays considered a therapeutic procedure, however, during ERCP several diagnostic tools can be used—brush cytology, endobiliary biopsy, and cholangioscopy. Brush cytology is an easy procedure, routinely performed in suspicious biliary strictures, characterized by low sensitivity (ranges 20–40%) and high specificity (ranges 92–100%) [6]. The sensibility can be improved by performing multiple brushings during the same ERCP and performing fluorescence in situ hybridization (FISH) analysis on the sample [9,10]. Endobiliary forceps biopsy allows the collection of a more complete tissue sample, including the subepithelial stroma, however, it is less commonly performed and technically more complex than brush cytology. This procedure has a sensitivity ranging between 43% and 81% [11]. The combination of brush cytology and forceps biopsy increases the sensitivity to 59.4% [12]. Both brush cytology and forceps biopsy are burdened by a low negative predictive value, thus when the report is negative for cancer but the radiological and clinical suspicious is high, other techniques should be applied to obtain a proper pathological analysis. Peroral cholangioscopy is a technique allowing direct visualization of the biliary mucosa and consequently a visual interpretation of the biliary stricture (Figure 1A). There are three types of cholangioscopy systems—mother and baby, single operator, and direct peroral cholangioscopy. In a meta-analysis, de Oliveira et al. reported that cholangioscopy has a sensitivity of 94% and a specificity of 95% [13]. Recently, the Monaco classification, in order to increase the diagnostic accuracy, identified eight visual parameters for biliary evaluation during cholangioscopy—the presence of stricture, lesion type, mucosal features, papillary projections, ulceration, abnormal vessels, scarring, and pronounced pit pattern [14]. Moreover, the direct visualization of a suspect area allows the execution of a targeted forceps biopsy increasing the accuracy of malignancy detection (Figure 1B) [15]. However, cholangioscopy devices are burdened by high costs and their widespread is consequently limited.

Echoendoscopy (EUS) is another endoscopic technique that can be used to evaluate a biliary stricture. EUS allows the visualization of the primary lesion and the locoregional staging evaluating the infiltration of adjacent tissues, lymph nodes, and vessel involvement [16]. The diagnostic accuracy of EUS has been shown to be higher in distal biliary tract strictures than in hilar strictures [17]. EUS fine-needle aspiration (EUS-FNA) is a well-established procedure for cytological sample collection, however, its application in hilar cholangiocarcinoma is not widely performed because of technical complexity [17]. In a meta-analysis involving 957 patients, the sensitivity and specificity of EUS-FNA in biliary stricture were respectively 80% and 97%. The sensitivity of EUS-FNA in proximal stricture was significantly lower than in distal strictures (respectively 76% and 83%) [18]. EUS-FNA can be a useful tool in case of ERCP sample collection failure [19].

Ultrasound imaging can be used also inside the biliary tree performing an intraductal endoscopic ultrasound (IDUS). It consists of a tiny probe inserted into the biliary ducts in order to evaluate the presence of malignant stigmata on the biliary ducts’ walls. In a retrospective study, IDUS showed a sensitivity and specificity in malignancy detection of 93.2% and 89.5% [20]. Other authors documented a higher sensitivity of IDUS when compared with EUS [21].

Recently, two other techniques became available for a direct evaluation of the biliary walls—confocal laser endomicroscopy (CLE) and optical coherence tomography (OCT). CLE uses a low-power laser to create a magnification of the mucosal layers (Figure 2) [22].

During ERCP, a CLE probe can be advanced through the duodenoscope channel and inserted into the biliary tree. The classification of Miami and the Paris inflammatory criteria have been developed in order to distinguish a malignant from an inflammatory biliary stricture [23,24]. The sensitivity and specificity of CLE have been reported to be 90% and 72%, respectively [25]. Conversely, OCT uses an infrared-light providing cross-sectional images of tissue reflectance in order to obtain information on microscopical tissue architecture [26]. Volumetric laser endomicroscopy (VLE) is a newer OCT that allows us to obtain higher definition in vivo cross-sectional images of the biliary wall layers [27]. OCT increases the sensitivity and accuracy of malignancy detection when compared to brush cytology alone [28]. M. Arvanitakis et al. described the role of OCT during ERCP to assess the diagnosis of a biliary stricture using two OCT criteria for malignancy—the unstructured walls layers and the presence of neovascularization. They reported an increased diagnostic accuracy when standard techniques (e.g., brush cytology and forceps biopsy) are combined with OCT [28]. However, the role of OCT as a single diagnostic tool in biliary malignancy is not well defined yet, and its use and widespread are limited due to high costs.

Endoscopy can play also a role in the biomarkers’ evaluation allowing bile samples collection. Classically in cholangiocarcinoma carbohydrate antigen 19-9 (CA 19-9) is considered the most accurate serum biomarker. However, CA 19-9 level >100 UI/mL has a sensitivity of 53% and a specificity of 75–90% in detecting cholangiocarcinoma [29]. Moreover, CA 19-9 can be raised in other malignancies (e.g., pancreatic cancer) and in benign conditions (e.g., cholangitis and primitive sclerosing cholangitis) [30]. Some authors used bile samples collected during ERCP to perform a multi-omic analysis (both metabolomic and proteomic), obtaining a panel of lipids and proteins that can discriminate patients with bilio-pancreatic malignancy [31]. Moreover, the role of extracellular vesicles (EVs) is raising for cancer detection. EVs concentration in bile collected during ERCP showed the capability to distinguish patients with malignancy from patients with benign biliary strictures with a higher level of accuracy when compared with EVs concentration in serum [32].

## 3. Indication for Biliary Drainage in Malignant HBO

The choice for biliary drainage is a complex assessment that should be taken by a multidisciplinary team. The first rule to keep in mind is that it is essential to complete the radiological abdominal staging, as the placement of a device into the biliary tree can interfere with the abdominal cross-sectional imaging (e.g., CT and MRI) [33]. Hence, the patients can be divided into two main groups—those who are eligible for a resection surgery and those requiring palliation therapy.

In resectable malignant HBO, the preoperative biliary drainage (PBD) is not routinely performed. Several retrospective studies showed that PBD increases the risk for post-surgical infections without any effect on survival [34,35,36]. In a systematic review and meta-analysis including 501 patients who underwent PBD and 391 patients who had not PBD, Celotti A. et al. showed that the two groups did not differ in terms of mortality rate but in terms of morbidity with increased risk for infective complications in patients undergoing PBD [37]. Scheufele et al. demonstrated that PBD induces a shift of the biliary microbiome with an increase of aggressive and resistant bacteria [38]. Given this background, the indication for PBD should be done balancing risks and benefits for each patient. In those undergoing left hepatectomy, PBD is not indicated as it increases the mortality rate mainly for the occurrence of post-operative sepsis [34]. Differently, one of the main causes of death after right hepatectomy is liver failure, and it has been shown that it is significantly more frequent in patients who did not undergo PBD [34]. This difference can be attributed to the higher volume of parenchyma loss in the right hepatectomy when compared to the left one, therefore, the quantification of future liver remnant (FLR) volume is essential to indicate whether PBD should be performed [39]. When the FLR volume is less than 30%, portal vein embolization (PVE) is required to obtain hypertrophy of the remnant liver; in this setting, PBD appears to reduce the risk for hepatic insufficiency and should be definitely performed [40]. Moreover, there is consensus that PBD is indicated in patients with cholangitis, hyperbilirubinemia-induced malnutrition, hepatic insufficiency or renal insufficiency, patients needing neo-adjuvant therapy, severely symptomatic patients, and those with delays in surgery [33].

The prognosis of cholangiocarcinoma is still today poor. Surgery is the only curative approach but it is feasible in just 30–40% of patients [5]. Criteria for non-resectability are distant metastases, lymph node metastases beyond the hepatoduodenal ligament, the bilateral ductal extension to the secondary (or sectorial) biliary branches, encasement or occlusion of the main portal vein (or common hepatic artery) proximal to its bifurcation, unilateral involvement of secondary (or segmental) biliary radicles with contralateral vascular involvement, lobar atrophy with the involvement of contralateral secondary (or sectorial) biliary radicles, and lobar atrophy with the involvement of contralateral portal vein or hepatic artery [41]. Biliary drainage in unresectable HBO represents the cornerstone for palliation. The aims of palliative biliary drainage are to enable chemotherapy and radiotherapy administration and improve the quality of life, relieving jaundice, pruritus, pain, and cholangitis [4].

## 4. Percentage of Liver Volume to Drain

Once assessed the indication for biliary drainage, the following step is to define how much of the liver parenchyma should be drained to relieve jaundice and reduce the risk of cholangitis.

In a retrospective study including 107 patients undergoing endoscopic stent placement for malignant HBO, Vienne A. et al. showed that draining >50% of the liver volume was a predictor of drainage effectiveness, particularly in Bismuth III stricture, and was associated with longer overall survival [42]. Interestingly, Takahashi E. et al. correlated the percentage of liver volume to drain with the patient’s liver function. They concluded that effective biliary drainage is obtained when >33% of the liver volume is drained in patients with preserved liver function and >50% in those with impaired liver function [43].

A cross-sectional study (e.g., CT or MRI) before performing the biliary drainage is crucial to define which liver sector will be drained in order to avoid opacification of undrained biliary ducts thus reducing the risk of post-procedural cholangitis [44]. Some authors suggested contrast-free cannulation to reduce the risk of post-ERCP cholangitis [45]. This technique is based on cross-sectional imaging as a guide-map to perform the cannulation of the obstructed duct injecting the contrast medium only when the obstruction is crossed thus opacificating only the drained sectors. Some authors described the use of other contrast media (e.g., air or CO_2_) to reduce the risk of infection [46].

Moreover, the cross-sectional imaging allows identifying the presence of portal vein thrombosis with consequent segmental parenchymal atrophy. Drainage of an atrophic liver segment should be avoided because it has been shown that it does not improve liver function, drainage effectiveness nor survival (Figure 3) [47]. On the contrary, it may increase the risk of cholangitis [42].

## 5. PTBD Versus ERCP for Biliary Drainage

In pre-operative HBO drainage, when indicated, the choice between ERCP and percutaneous approach is not standardized. In a meta-analysis of 15 studies, Hameed A. et al. [48] showed that the occurrence of liver failure was higher in patients undergoing percutaneous transhepatic biliary drainage (PTBD) when compared with endoscopic biliary drainage (EBD, including endoscopic nasobiliary drainage or stent placement). Moreover, the median one-year and five-year survival was higher in EBD (respectively 91% versus 73% and 46% versus 30%). The incidence of procedure-related complications, such as cholangitis and pancreatitis, was not statistically different between the two groups, however, there was a trend of fewer complications in PTBD patients. In another meta-analysis, Mahjoub A. et al. reported that the overall procedure-related complications rate (in terms of cholangitis and pancreatitis occurrence) was lower in the PTBD than in the EBD group. However, post-operative morbidity and mortality were higher in the PTBD group (26% versus 21% and 7.5% versus 3.8%, respectively) with a trend towards better outcomes in the EBD group although not statistically significant. In palliative biliary drainage, ERCP is generally preferred rather than PTBD [44]. Moreover, biliary drainage via ERCP showed lower adverse events rate and shorter hospitalization when compared with PTBD [49]. In a propensity score matching analysis, Komaya K. et al. reported that patients undergoing PTBD have lower overall survival and a higher risk for seeding metastasis when compared with ERCP [50].

The decision to perform biliary drainage via ERCP or PTBD is complex and may be different among centers depending on technical experience and local facilities. Generally, PTBD is preferred when the patient presents gastro-duodenal altered anatomy, when the bile ducts to drain are not accessible by ERCP, and when ERCP was not sufficient to achieve adequate biliary drainage. The major drawback of PTBD is the need for an external catheter that may be a route of infection, it may represent discomfort for the patient causing local pain, aesthetic inconvenience worsening the patient’s quality of life (Figure 4) [51].

PTBD and ERCP should not be considered as exclusive—in case of complex HBO, which is difficult to approach by ERCP, initially, a PTBD can be performed and the PTBD tube can be used as a guide to place a stent via ERCP (Rendez-vous technique) [52].

## 6. ERCP for Biliary Drainage

Endoscopic drainage in malignant HBO is challenging and should be performed in high volume centers [44]. In a meta-analysis involving 13 studies, Keswani R. et al. showed a higher success rate and lower occurrence of adverse events when ERCPs are performed in high volume centers particularly for advanced procedures [53]. To perform biliary drainage, three options are available—plastic stent (PS), nasobiliary drainage, and self-expandable metal stent (SEMS).

Plastic stents are indicated in pre-operative drainage and when the treatment approach (curative versus palliative) has not yet been defined. The advantages of PS are their removability, thus not prohibiting further therapeutic approaches, and their moldability. Therefore, the caliber and the length of PS can be adapted to the singular biliary tree shape. When sufficient biliary drainage is not achieved, multiple PS can be inserted in order to increase the biliary tree patency (Figure 5A,B).

Major drawbacks of PS are the risk of stent migration and stent occlusion [53,54]. PS migration occurs in 5–10% of cases, with the distal migration more common when compared with the proximal migration [54]. In a large retrospective study, Arhan M. et al. documented a lower incidence of stent migration in malignant biliary stricture and multiple stent placement when compared respectively with benign biliary stricture and single or double PS placement [55]. Stent occlusion occurs in up to 30% of cases, and it is related to bacterial biofilm formation, biliary sludge, biliary reflux of dietary fibers, and clots formation [55,56]. Stent patency mainly depends on stent caliber—larger PS has longer patency and time of placement, and PS exchange is generally needed every 3–6 months [57].

Endoscopic nasobiliary drainage (ENBD) can be an option in pre-operative malignant HBO drainage [58]. In a meta-analysis involving 925 patients with malignant biliary obstruction, Lin H. et al. reported a lower rate of pre-operative cholangitis, post-operative fistula, and stent dysfunction in patients undergoing ENBD compared with EBD [59]. Nasobiliary tube occlusion in hilar cholangiocarcinoma has been reported to occur more rarely than in plastic stents [60]. ENBD enables us to perform a cholangiogram injecting contrast medium through the drainage tube without any other invasive procedures, thus helping in the diagnosis and management of complications. However, the major disadvantage of ENBD is the patient’s discomfort and consequently its short-term usability.

SEMS is indicated in palliative drainage. Several studies showed the superiority of SEMS in palliation of unresectable malignant HBO when compared with PS [61]. In a meta-analysis, Sawas T. et al. documented that SEMS is associated with a lower risk of short-term and long-term stent occlusion, lower incidence of therapeutic failure, lower cholangitis occurrence rate, and less need for reinterventions when compared with PS [62]. Moreover, SEMS was associated with longer overall survival when compared with PS [63]. Different kinds of SEMS are nowadays available—fully-covered SEMS (FC-SEMS), partially-covered SEMS (PC-SEMS), and uncovered-SEMS (U-SEMS). In malignant HBO, the use of U-SEMS is suggested because the uncovered mash enable the side biliary branches drainage [64]. In a retrospective study involving 30 patients, Inoue T. et al. reported the use of FC-SEMS in malignant HBO—although technically feasible, the occurrence of hepatic abscess in 7% of patients burden the risk of intrahepatic bile duct occlusion with consequent septic complications [65]. Kitamura et al. described the use of PC-SEMS in malignant HBO as an alternative to U-SEMS in order to reduce the risk for tumor ingrowth and to maintain the possibility of stent removal [66]. However, data on FC-SEMS and PC-SEMS in malignant HBO are still inconsistent and only U-SEMS are indicated in this setting for palliation (Figure 5B,C). Major drawbacks of U-SEMS are the non-removability and the difficult management of stent obstruction [67]. U-SEMS can occlude for the presence of biliary sludge or of tumor ingrowth—in the first case, the management is similar to biliary lithiasis (Figure 6); the second case is more challenging and may require the placement of a second SEMS or PS.

## 7. Complete Versus Incomplete Drainage

Several studies investigated whether the biliary drainage should be unilateral or bilateral [68,69,70]. On one hand, the unilateral drainage raises the issue of insufficient jaundice relief and risk of infective complications occurrence, on the other hand, bilateral stenting is burden by a higher technical complexity. In a multicenter, prospective, randomized study unilateral and bilateral endoscopic drainage showed similar results in terms of success rate, but unilateral stenting was associated with a higher risk for reintervention and shorter patency time [71]. In a recent meta-analysis of 21 studies involving 1292 patients with malignant HBO, Meybodi M. et al. reported that the technical success and the functional success rate were higher in unilateral drainage when compared with the bilateral [70]. Short-term and long-term complication rates were comparable in the two groups [70]. In another meta-analysis, bilateral stenting, considering both SEMS and PS, has been shown to be more effective in lowering hyperbilirubinemia [69]. Moreover, bilateral SEMS seemed to be associated with a lower incidence of complications, while bilateral PS had similar odds of complications when compared with unilateral drainage [69]. A major limitation of these studies is the definition of unilateral and bilateral, respectively when one or two stents are placed in the biliary tree. This seems to be a simplification considering the great complexity of malignant HBO. Therefore, in Bismuth I, the placement of one stent is enough to obtain the complete liver drainage; in Bismuth II, the placement of one stent can drain up to 50% of the liver (right or left liver), while two stents will drain all the liver; in Bismuth III and IV, even two stents may not be enough to obtain sufficient drainage (Figure 7). Thus, the concept of unilateral and bilateral should be replaced by complete drainage and incomplete drainage [64].

When multiple SEMSs have to be placed in the biliary tree, two different techniques are available: stent-in-stent (SIS) and side-by-side (SBS). The SIS method consists of placing the second SEMS through the first SEMS, and the SBS techniques involve the parallel placement of multiple SEMS simultaneously or sequentially. The studies comparing the two techniques are scanty. On one hand, some authors documented an increased rate of adverse events (e.g., cholangitis and liver abscess) and longer stent patency in SBS when compared to SIS [72]. Conversely, other authors described prolonged patency in SIS than in SBS [73]. Finally, some others found no differences in complication rate, patency, and overall survival comparing the two techniques [74]. The SBS method is generally preferred because the deployment of multiple stents is technically easier than in SIS and, more importantly because in case of stent malfunctioning (e.g., stent occlusion for tumor ingrowth) the reintervention is usually possible and successful unlike in SIS in which retreatment may be prohibiting [75].

## 8. EUS-Guided Biliary Drainage

Recently, EUS has emerged as an option in biliary drainage (echoendoscopic biliary drainage (EUS-BD)), particularly when transpapillary ERCP drainage has failed [76]. Two different routes can be used to access the biliary tree with EUS—the intrahepatic and the extrahepatic bile duct approaches. The extrahepatic bile duct puncturing is valuable for distal biliary obstruction, thus for HBO, the intrahepatic approach is needed [77]. The EUS-BD has three possible drainage routes—the transmural placement of a stent creating a novel biliodigestive anastomosis; the transpapillary antegrade technique that involves the dilatation of the puncture site, the passage of a guidewire through the stenosis until the papilla, and the releasing of a stent antedradelly; the transpapillary retrograde technique (Rendez-vous technique), requiring the insertion of a guidewire through the stenosis until the papilla, the exchange of the instrument and the placement of a stent via the papilla using the guidewire as a route [77], and the transgastric drainage. In malignant HBO, the EUS-BD can be considered as an alternative to PTBD in case of altered upper gastrointestinal anatomy, duodenal obstruction, gastric outlet obstruction, periampullary diverticulum, distal biliary tumor infiltration, or occluded biliary metal stent [78].

In a meta-analysis, Baniya R. et al. showed no statistically significant differences in terms of technical and clinical success rate between PTBD and EUS-BD in malignant biliary obstruction with a lower incidence of moderate-severe adverse events in the EUS-BD group [79]. Among the different methods available, the hepaticogastrostomy is effective for left-sided biliary decompression [80]. The hepaticogastrostomy involves the puncture of the left intrahepatic bile duct using the transgastric EUS imaging in order to deliver a stent between the bile tree and the stomach. This technique has a technical success rate of 91–100% and a clinical success rate of 75–100% [81]. The incidence of adverse events has been reported to be 25% including stent migration, bile leaks, pneumoperitoneum, and cholangitis [81,82]. Ogura T. et al. described the “bridging method” to approach the right biliary system. This requires the puncture of the left intrahepatic biliary duct through the stomach, the advance of a guidewire through the right intrahepatic biliary duct, the delivery of a metal stent between the right and the left intrahepatic ducts, and finally, the performance of the hepaticogastrostomy [83].

The role of EUS-BD has been described also as a rescue reintervention in patients presenting metallic stent dysfunction in which an ERCP attempt had failed [84]. EUS-BD can also be combined with ERCP to provide complete drainage. When a SEMS is deployed in the left biliary tract a concomitant EUS-hepaticoduodenostomy can be performed, conversely, when a SEMS is placed in the right hepatic duct, a EUS-hepaticogastrostomy can be used to complete the drainage [85].

Once the biliary tree has been accessed, the choice between the transmural or transpapillary drainage is not standardized. The transpapillary drainage is generally more complex compared with the transmural, in fact, it requires the antegrade placement of a guidewire, access to the papilla in the Rendez-vous method, and the dilatation of the puncture tract in the antegrade method [77]. Moreover, the transmural drainage provides easy access to the biliary tree in case of reintervention.

These different techniques for EUS-BD should not be considered exclusive, but they can be considered complementary and can be chosen to provide the best drainage for personalized treatment of every single patient.

## 9. Loco-Regional Therapies

The only curative approach of cholangiocarcinoma is surgery, however, only a small percentage of patients can be referred to surgery because of locally advanced cancer or the presence of distant metastases. In order to prolong the overall survival and to improve the quality of life in patients with unresectable hilar cholangiocarcinoma, locoregional techniques have been developed such as photodynamic therapy (PDT), radiofrequency ablation (RFA), and brachytherapy (BT). PDT is a minimally invasive procedure that has been described as a palliative approach in advanced cholangiocarcinoma [86]. PDT involves the injection of a photosensitizer followed by irradiation with a specific wavelength in order to produce selective cytotoxicity on cancer cells [87]. PDT in cholangiocarcinoma can be performed both via ERCP and percutaneous transhepatic cholangioscopy (PTCS).

During ERCP, after injecting the photosensitizer, an optical fiber is inserted into the strictured bile ducts, and a light is applied in order to obtain the oxygen free radical formation within the tumor cells [88]. In cholangiocarcinoma, PDT has been shown to reduce malignant biliary stenosis and to be an option in post-surgical recurrence [88,89]. In a randomized prospective study, Ortner M. et al. reported a longer overall survival in patients with unresectable cholangiocarcinoma undergoing PDT when compared with standard treatment [90]. Li Z. et al. documented that patients with hilar HBO undergoing stent placement and PDT (both via ERCP and PTCS) had a significantly longer median survival, improved quality of life, and no differences in post-operative adverse events occurrences when compared with patients undergoing only stents placement [91]. In a retrospective study, PDT via ERCP and PTCS were compared and showed no significant differences in terms of overall survival and median metal stent patency [92].

RFA is another local therapy used in several solid tumors, which involves the production of high temperatures inside the tumor leading to tissue necrosis and consequently reduction in tumor size [93]. RFA can be performed percutaneously, intra-operatively, or endoscopically. During ERCP, intrabiliary-RFA is performed placing an RFA catheter under fluoroscopic guidance through the malignant biliary stenosis and releasing thermic energy for a standardized amount of time (Figure 8) [94].

In unresectable cholangiocarcinoma, intrabiliary-RFA is considered an option prior to SEMS placement in order to prolong the sent patency and the overall survival [95]. Intrabiliry-RFA has also been applied as an ablative treatment for SEMS occlusion for tumor ingrowth [96]. In a meta-analysis involving 505 patients with unresectable biliary stricture, Sofi A. et al. reported a significant longer stent patency and overall survival in patients receiving RFA when compared with those treated only with stent placement [97]. Moreover, the risk for adverse events was not higher in the RFA group except for post-procedural abdominal pain [97].

PDT and RFA in hilar HBO can have a role in prolonging stent patency and consequently in improving quality of life, reducing reintervention for stent occlusion, and increasing overall survival, however, randomized studies are needed to clarify their role in daily clinical practice.

Another option for locoregional treatment in hilar cholangiocarcinoma is BT. Cholangiocarcinoma has been shown to be responsive to radiotherapy which is mainly used as an adjuvant, neoadjuvant, or palliative treatment [4]. The application of external beam radiotherapy in hilar tumors can be challenging for the risk of damage of surrounding organs, thus BT enables the local delivery of high dose radiation reducing the exposure to radiation of adjacent tissues. In biliary malignancy, BT can be performed percutaneously or endoscopically. The endoscopic approach involves the placement of a nasobiliary tube (NBT) through the biliary stricture and the insertion of a BT catheter inside the NBT for the delivery of high dose radiation (Figure 9) [98].

## 10. Conclusions

Malignant HBO is a complex scenario that needs a multidisciplinary approach from diagnosis to final treatment. Each clinical case should be considered unique and should be discussed by radiologists, oncologists, surgeons, and endoscopists in order to personalize the care pathway in terms of diagnosis, treatment, and palliation.

The diagnostic algorithm (Figure 10) in patients with HBO should always start with a clinical and laboratory assessment, including evaluation of liver function and biochemical markers (e.g., CA 19-9 and CEA). Cross-sectional imaging (e.g., CT or MRCP) is considered mandatory and should be obtained before any interventional approach in order to perform a radiological diagnosis, staging, and evaluation of resectability. In the case of an HBO with radiological stigmata of malignancy fulfilling the criteria for resectability, the patient can be referred to surgery also without a pathological sample. Otherwise, in the case of unresectable malignant-HBO, a cytological or histological sample is mandatory. It can be obtained during ERCP by performing brush cytology or forceps biopsy. However, because of the low negative predictive value of these techniques, in case of a negative report for cancer but with a high radiological suspect, another cytological or histological sample should be obtained. In this setting, an ERCP guided brush cytology or forceps biopsy can be reattempted, or another technique can be used (e.g., EUS-FNA, peroral cholangioscopy, OCT).

Once the diagnostic and staging pathway has been completed, the therapeutic algorithm can be applied (Figure 11). Commonly, the patient with malignant-HBO presents obstructive jaundice. In this setting, whether to perform a biliary drainage and which technique to use should be carefully evaluated in the singular patient. In resectable HBO candidates to left hepatectomy, the biliary drainage is generally not required. In candidates to right hepatectomy, the biliary drainage is required as a bridge to surgery in case of an FLR <30% requiring a PVE. In this case a PTBD, a PS, or nasobiliary drainage (NBD) can be used. In unresectable HBO, biliary drainage represents the standard of care. Nowadays, endoscopic drainage is generally preferred over the percutaneous approach, however, the technique used can vary based on local facilities and experience. When ERCP is preferred for palliation, a U-SEMS should be considered the first choice. In case of failure or technical complexity (e.g., altered anatomy) a EUS-BD can be attempted.

Regardless of the technique used, the biliary drainage should be as complete as possible (> 50% of live volume should be drained) in order to reduce the risk for infection and the risk for liver failure.

The application of new locoregional techniques (e.g., PDT, RFA, BT) can be considered to control the locoregional growth. However, their applicability in daily clinical practice is not still standardized and more clinical trials are needed to assess their use. Malignant HBO is such a complex condition that a standardized endoscopic approach may be challenging. Each patient should be carefully analyzed in order to define the pro and cons of the several endoscopic procedures available. Therefore, a personalized endoscopic approach is mandatory with the aim of treating the patient in its complexity and uniqueness.

## Figures and Tables

**Figure 1 jpm-11-00078-f001:**
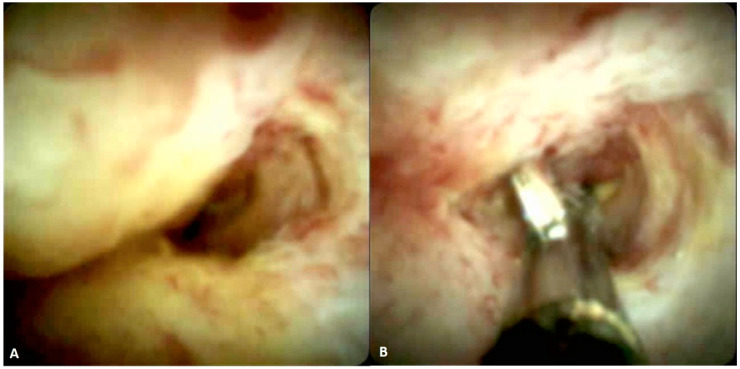
Peroral cholangioscopy of (**A**) a malignant hilar stricture and (**B**) target biopsy under direct visualization.

**Figure 2 jpm-11-00078-f002:**
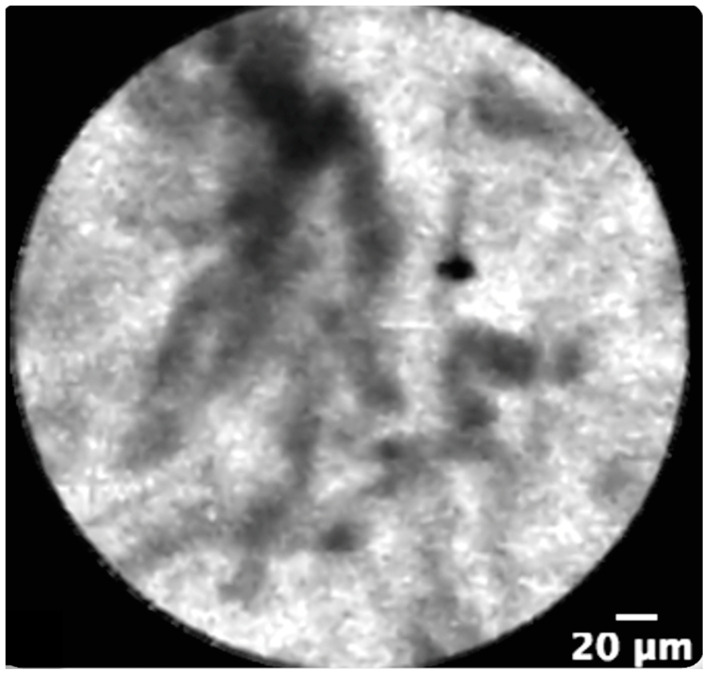
Confocal laser endomicroscopy showing a thickened reticular structure representing a criterion for malignancy.

**Figure 3 jpm-11-00078-f003:**
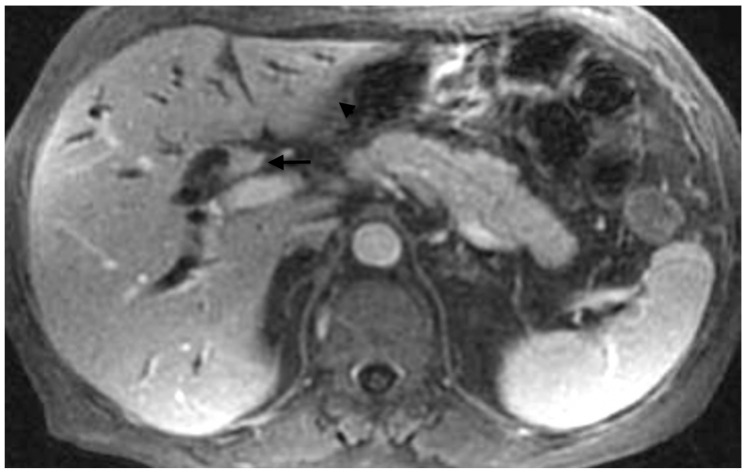
Abdominal MRI showing a hilar cholangiocarcinoma (arrow) with left liver lobe atrophy (arrowhead).

**Figure 4 jpm-11-00078-f004:**
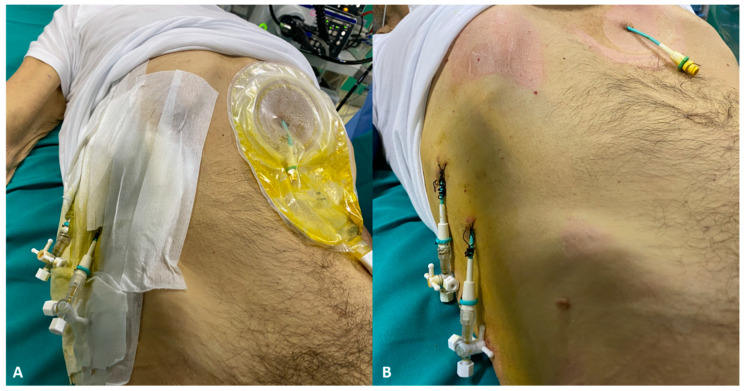
Three percutaneous drainages in a patient with hilar cholangiocarcinoma. (**A**) Bile spillage from the drainages (**B**) with skin erythema.

**Figure 5 jpm-11-00078-f005:**
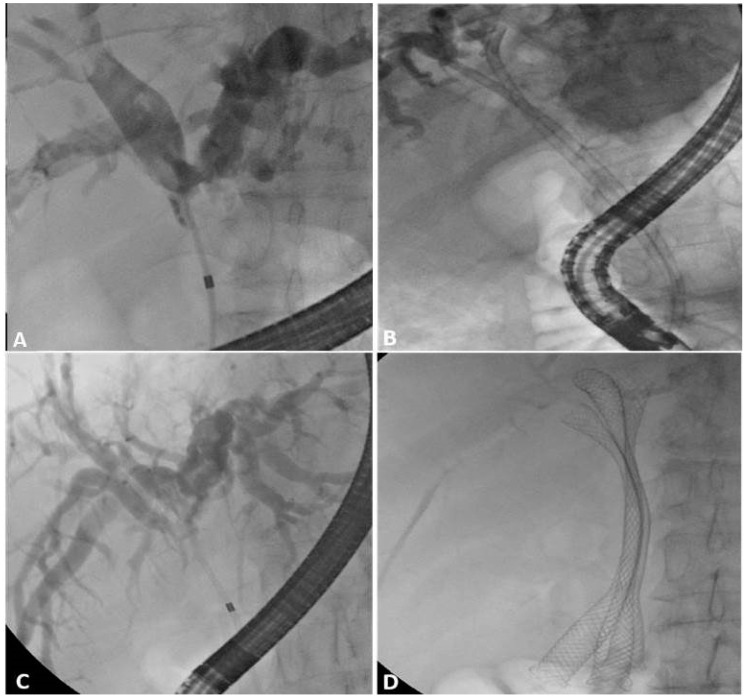
(**A**) Bismuth type II malignant hilar stricture (**B**) drained with two plastic stents. (**C**) Bismuth type IIIa malignant hilar stricture (**D**) drained with three self-expandable metal stents.

**Figure 6 jpm-11-00078-f006:**
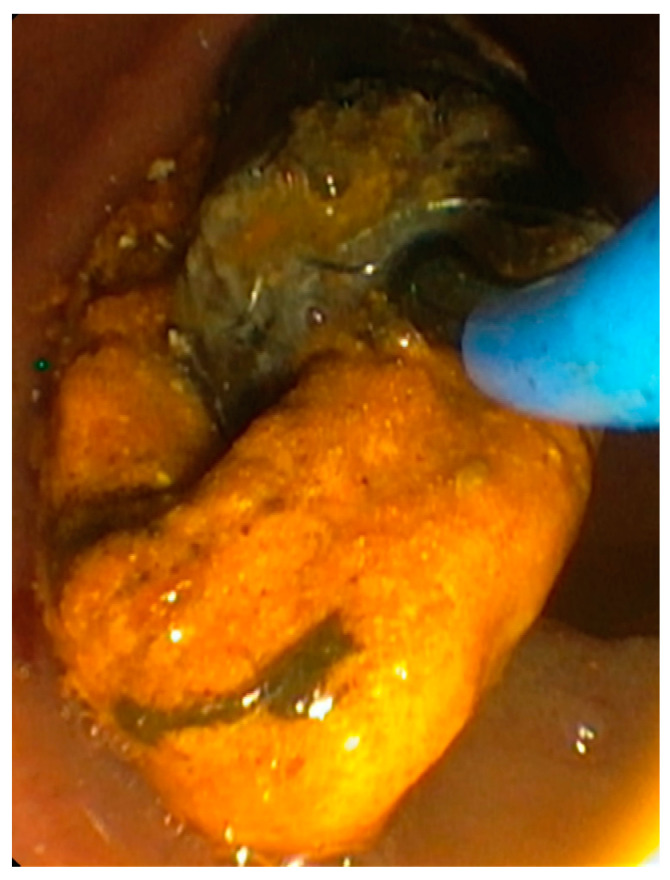
Endoscopic intervention for occluded self-expandable metal stent; sludge removal with a Fogarty balloon.

**Figure 7 jpm-11-00078-f007:**
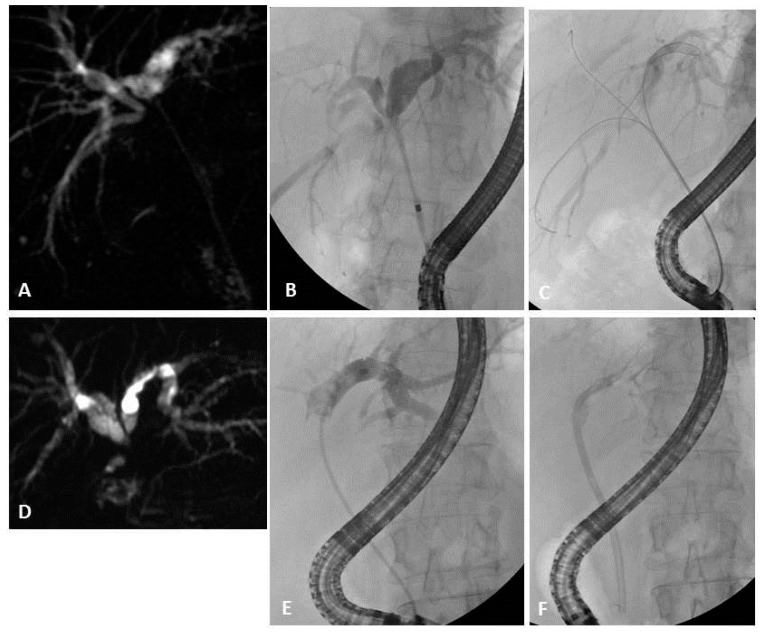
Magnetic resonance cholangiography (MRC) showing (**A**) Bismuth type IIIa malignant hilar stricture (**B**) confirmed by Endoscopic Retrograde Cholangio-Pancreatography (ERCP); (**C**) three guidewires are placed and (**D**) three plastic stents are inserted, obtaining complete drainage of all the liver segments. (**E**) MRC of a Bismuth type II malignant hilar stricture; (**F**) during ERCP, only the left hepatic ducts are opacified and a single plastic stent is inserted, obtaining incomplete drainage of the liver.

**Figure 8 jpm-11-00078-f008:**
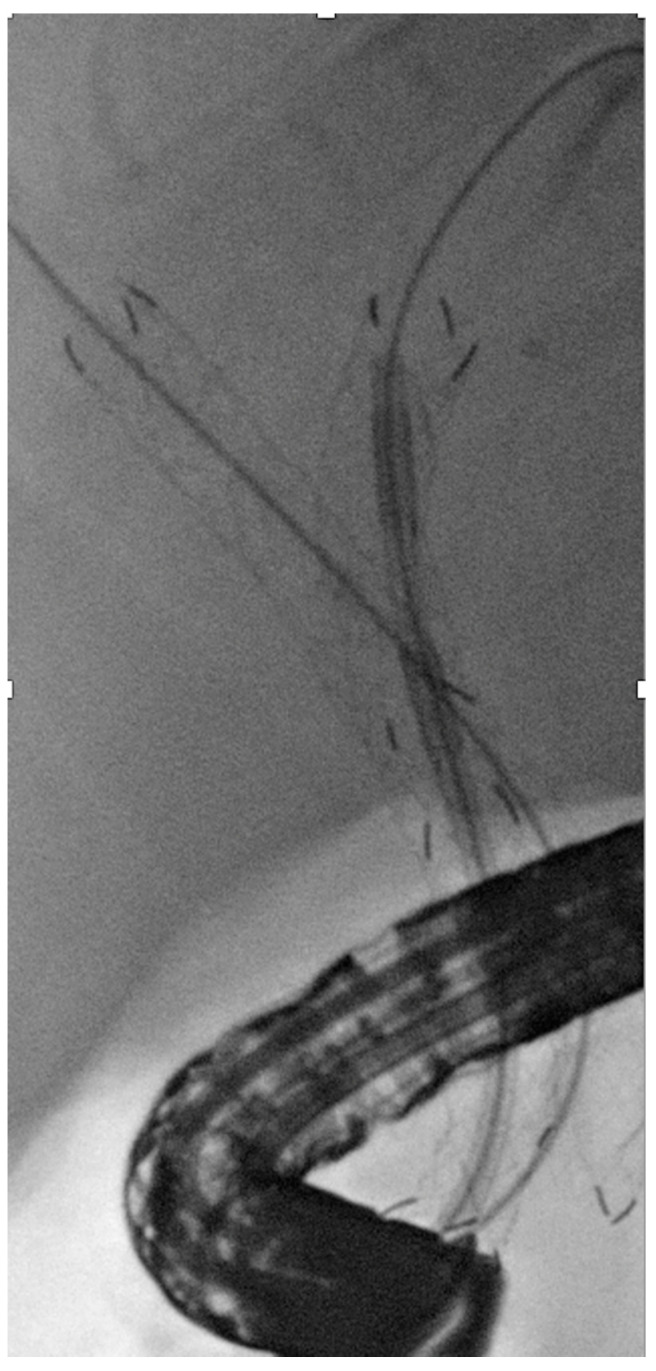
Radiofrequency ablation of hilar cholangiocarcinoma after self-expandable metal stents placement.

**Figure 9 jpm-11-00078-f009:**
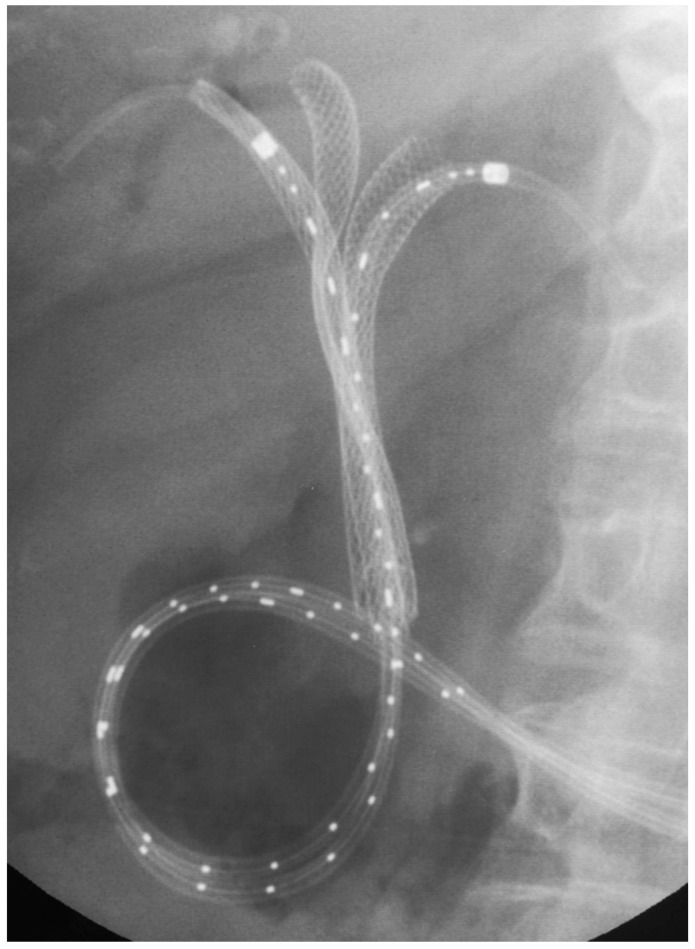
Three self-expandable metal stents with two nasobiliary drains with catheters for intraluminal brachytherapy in a case of hilar cholangiocarcinoma.

**Figure 10 jpm-11-00078-f010:**
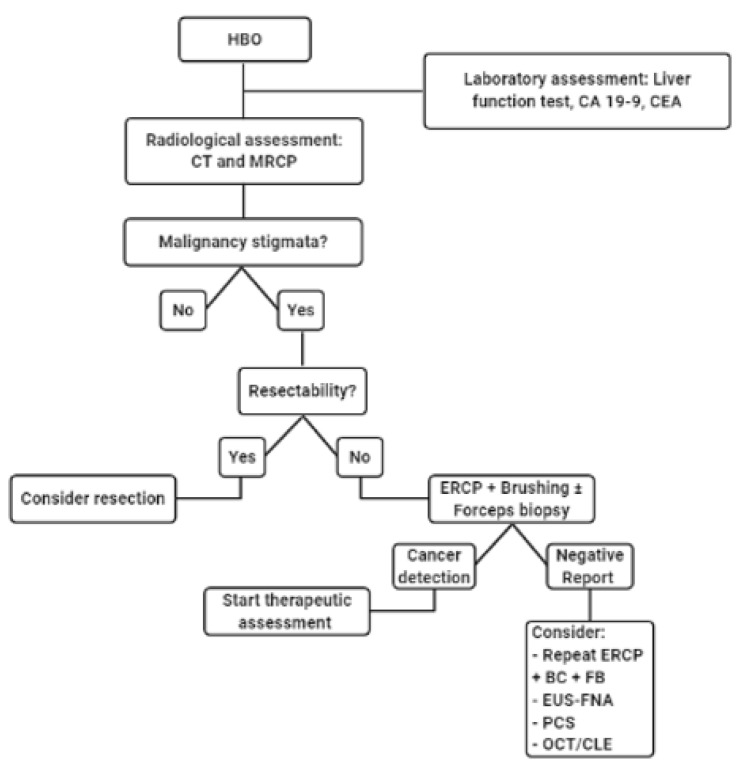
Diagnostic algorithm (HBO: hilar biliary obstruction; Ca 19-9: carbohydrate antigen 19-9; CEA: carcinoembryonic antigen; CT: computed tomography; MRCP: magnetic resonance cholangiopancreatography; ERCP: endoscopic retrograde cholangiopancreatography; BC: biliary brushing; FB: endoscopic forceps biopsy; EUS-FNA: echoendoscopic fine-needle aspiration; PCS: peroral cholangioscopy; OCT: optical coherence tomography; CLE: confocal laser endomicroscopy).

**Figure 11 jpm-11-00078-f011:**
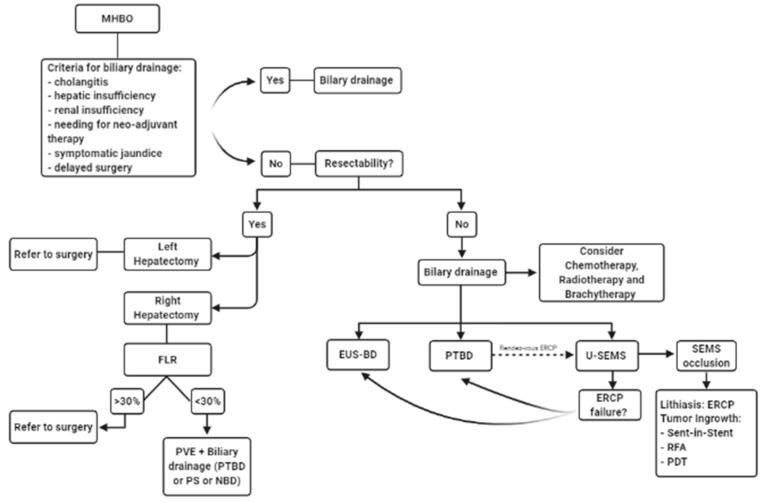
Therapeutic algorithm (MHBO: malignant hilar biliary obstruction; FLR: future liver remnant; PVE: portal vein embolization; PTBD: percutaneous transhepatic biliary drainage; PS: plastic stent; NBD: nasobiliary drainage; EUS-BD: echoendoscopic biliary drainage; U-SEMS: uncovered self-expandable metal stent; SEMS: self-expandable metal stent; ERCP: endoscopic retrograde cholangiopancreatography; RFA: radiofrequency ablation; PDT: photodynamic therapy).

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
