# Peer review of "Personalized Endoscopy in Complex Malignant Hilar Biliary Strictures"

_jpm, 2021, doi:10.3390/jpm11020078_

Round 1

Reviewer 1 Report

This is a well-written review of management of hilar MBO. 

  1. Page 2. The authors suggested Figure 1 for Bismuth Classification but no figure attached. Please add a figure.
  2. Page 5. OCT is not well known compared to other modalities. Please add data on its diagnostic yield.
  3. Figure 3. This is MRI. not CT. The picture should be replaced by CT with PV thrombus.
  4. Page 7. Comparison of ERCP and PTBD. PTBD has a potential risk of tumor seeding and might be oncologically inferior (Surgery. 2017;161(2):394-404.).
  5. Page 11. Bilateral vs. unilateral SEMS. A RCT of bi- and unilateral SEMS should be cited (Gastrointest Endosc. 2017;86:817-827.).
  6. Page 13. Please give some comments on combined ERCP and EUS-BD (Dig Endosc. 2019 Apr;31 Suppl 1:50-54.).

Author Response

First reviewer comments:

  • Comment: Page 2. The authors suggested Figure 1 for Bismuth Classification but no figure attached. Please add a figure.

Answer: thank you for the comment, we have erroneously inserted the term “Figure 1” in the wrong line. We provided to delete it.

  • Comment: Page 5. OCT is not well known compared to other modalities. Please add data on its diagnostic yield.

Answer: Thank you for your interesting comment. We perfectly agree that the role of OCT in the diagnosis of biliary malignancy is not well defined yet, particularly when used as single diagnostic tool. In lines 142-146 we added the description of a study that investigates the role con ERCP and OCT combination to increase the diagnostic accuracy of biliary malignancy. In line 146-147 we underline that the use of OCT in the diagnosis of biliary malignancy is still not defined and that its widespread is limited also by high costs.

  • Comment: Figure 3. This is MRI. not CT. The picture should be replaced by CT with PV thrombus.

Answer: we mistakenly wrote the term “CT-scan” instead of “Abdominal-MRI”. We provided the correct picture legend with the proper description of the MRI showing an hilar cholangiocarcinoma with left liver lobe atrophy.

  • Comment: Page 7. Comparison of ERCP and PTBD. PTBD has a potential risk of tumor seeding and might be oncologically inferior (Surgery. 2017;161(2):394-404.).
  • Answer: we agree with the reviewer that data have been published about the risk for tumor seeding with PTBD. We added, in line 244-246, the description of the study citated by the reviewer underling that the increased risk of seed metastasis reduces the overall survival in patient undergoing PTBD when compared with those undergoing ERCP. We added the reference in line 693. 

  • Comment: Page 11. Bilateral vs. unilateral SEMS. A RCT of bi- and unilateral SEMS should be cited (Gastrointest Endosc. 2017;86:817-827.).
  • Answer: thank you for the comment. We added the reference as requested by the reviewer in line 755. Moreover, in lines 326-328 we reported that this multicenter, prospective, randomized study documented the superiority of bilateral stenting in patency duration and lower reintervention rate.

  • Comment: Page 13. Please give some comments on combined ERCP and EUS-BD (Dig Endosc. 2019 Apr;31 Suppl 1:50-54.).

Answer: thank you for the suggested reference. We added in lines 395-398 a description of the         potential role of combination therapy with EUS and ERCP when ERCP or EUS alone are not enough to ensure a complete drainage. We added the reference in line 797.

Reviewer 2 Report

Nice overview by Boskoski and colleagues about diagnostic and therapeutic options in perihiliar cholangiocarcinoma (CCA). I have only few concerns which should be addressed:

Diagnostic approach

1. EUS has high diagnostic accuracy in the distal DHC (e.g., strictures, duct stones) and less accuracy in the proximal DHC (MRCP preferred here)

2. A disadvantage of cholangioscopy are the relatively high costs for the devices (e.g., Spyscope)

Indication for biliary drainage

3. To the best of my Knowledge there are two clear indications for pre-operative biliary drainage in expected CCA:

  • Severe cholestasis (e.g., bilirubin > 12 mg/dl)
  • Cholangitis

Therapy in CCA

4. Liver transplantation is a therapeutic option in selected patients likewise.

The authors should comment these concerns.

Author Response

Second reviewer comments:

  • Comment: EUS has high diagnostic accuracy in the distal DHC (e.g., strictures, duct stones) and less accuracy in the proximal DHC (MRCP preferred here)

Answer: we agree with the reviewer that EUS has a high accuracy in distal biliary tract strictures. We underlined in line 116 this concept. Moreover, in 119-122 we reported that also the accuracy of EUS-FNA is lower in proximal strictures.

  • Comment: A disadvantage of cholangioscopy are the relatively high costs for the devices (e.g., Spyscope)
  • Answer: in line 108-109 we added the sentence “However, cholangioscopy devises are burdened by high costs and their widespread is consequently limited” to underline the high costs of cholangioscopy as requested by the reviewer.

  • Comment: To the best of my Knowledge there are two clear indications for pre-operative biliary drainage in expected CCA: Severe cholestasis (e.g., bilirubin > 12 mg/dl), Cholangitis

Answer: we perfectly agree with the reviewer that cholangitis and severe cholestasis are clear indications for biliary drainage. In the text we also included “hyperbilirubinaemia-induced malnutrition, hepatic insufficiency or renal insufficiency, patients needing neo-adjuvant therapy, severely symptomatic patients and those with delays in surgery” as possible indications for biliary drainage. Moreover, biliary drainage may be required in some cases when the FLR volume is less than 30% and portal vein embolization (PVE) is required to obtain hypertrophy of the remnant liver; in this setting the biliary drainage may reduce the risk for hepatic insufficiency.

  • Comment: Liver transplantation is a therapeutic option in selected patients likewise.
  • Answer: we thank the reviewer for the comment. We agree that liver transplantation is a therapeutic option in selected patients with cholangiocarcinoma, however our review focused on the endoscopic approach in malignant biliary stricture and some surgical and medical approach have not been investigated.